# A Handcrafted Radiomics-Based Model for the Diagnosis of Usual Interstitial Pneumonia in Patients with Idiopathic Pulmonary Fibrosis

**DOI:** 10.3390/jpm12030373

**Published:** 2022-02-28

**Authors:** Turkey Refaee, Benjamin Bondue, Gaetan Van Simaeys, Guangyao Wu, Chenggong Yan, Henry C. Woodruff, Serge Goldman, Philippe Lambin

**Affiliations:** 1The D-Lab, Department of Precision Medicine, GROW-School for Oncology, Maastricht University, 6200 MD Maastricht, The Netherlands; t.refaee@maastrichtuniversity.nl (T.R.); ycgycg007@gmail.com (C.Y.); h.woodruff@maastrichtuniverstiy.nl (H.C.W.); 2Department of Diagnostic Radiology, Faculty of Applied Medical Sciences, Jazan University, Jazan 45142, Saudi Arabia; 3Department of Pneumology, Erasme University Hospital, Université libre de Bruxelles, 1070 Brussels, Belgium; benjamin.bondue@erasme.ulb.ac.be; 4Department of Nuclear Medicine, Erasme University Hospital, Université libre de Bruxelles, 1070 Brussels, Belgium; Gaetan.van.simaeys@erasme.ulb.ac.be (G.V.S.); serge.goldman@ulb.ac.be (S.G.); 5Department of Radiology, Union Hospital, Tongji Medical College, Huazhong University of Science and Technology, Wuhan 430074, China; g.wu@maastrichtuniverstiy.nl; 6Department of Medical Imaging Center, Nanfang Hospital, Southern Medical University, Guangzhou 510515, China; 7Department of Radiology and Nuclear Medicine, Maastricht University Medical Centre+, 6200 MD Maastricht, The Netherlands

**Keywords:** handcrafted radiomics, interstitial lung diseases, usual interstitial pneumonia, machine learning

## Abstract

The most common idiopathic interstitial lung disease (ILD) is idiopathic pulmonary fibrosis (IPF). It can be identified by the presence of usual interstitial pneumonia (UIP) via high-resolution computed tomography (HRCT) or with the use of a lung biopsy. We hypothesized that a CT-based approach using handcrafted radiomics might be able to identify IPF patients with a radiological or histological UIP pattern from those with an ILD or normal lungs. A total of 328 patients from one center and two databases participated in this study. Each participant had their lungs automatically contoured and sectorized. The best radiomic features were selected for the random forest classifier and performance was assessed using the area under the receiver operator characteristics curve (AUC). A significant difference in the volume of the trachea was seen between a normal state, IPF, and non-IPF ILD. Between normal and fibrotic lungs, the AUC of the classification model was 1.0 in validation. When classifying between IPF with a typical HRCT UIP pattern and non-IPF ILD the AUC was 0.96 in validation. When classifying between IPF with UIP (radiological or biopsy-proved) and non-IPF ILD, an AUC of 0.66 was achieved in the testing dataset. Classification between normal, IPF/UIP, and other ILDs using radiomics could help discriminate between different types of ILDs via HRCT, which are hardly recognizable with visual assessments. Radiomic features could become a valuable tool for computer-aided decision-making in imaging, and reduce the need for unnecessary biopsies.

## 1. Introduction

Idiopathic pulmonary fibrosis (IPF) is the most common progressive form of interstitial lung disease (ILD) with an unknown etiology, usually impacting older adults [1,2]. In 2011, four societies—the American Thoracic Society, the European Respiratory Society, the Japanese Respiratory Society, and the Latin American Thoracic Association—came together to issue an evidence-based statement, which provided recommendations for both the diagnosis and management of IPF [3]. According to these recommendations, high-resolution computed tomography (HRCT) can play a crucial role in the diagnosis of fibrotic lung diseases and has a significant impact on medical decision-making.

Diagnosing IPF comes about using a multidisciplinary discussion (MDD) of the clinical, radiological, and, if available, pathological data showing a usual interstitial pneumonia (UIP) pattern which is the most common histopathological form of diffuse lung fibrosis [3,4]. The diagnostic radiological characteristic of UIP necessitates honeycombing with a basal and subpleural predominance. The upper lobes are less affected, and traction bronchiectasis may be present [5]. An IPF diagnosis requires a multidisciplinary discussion (MDD) and the exclusion of known causes of ILD, in addition to the presence of a UIP-specific pattern on thin-section CT, or a specific combination of HRCT patterns and histopathological UIP patterns in patients subjected to lung tissue sampling [3]. It is also worth noting that, in 2018, the Fleischner Society expanded on these recommendations for diagnosing IPF to include the appearance of probable UIP in HRCTs, if the clinical context was consistent with an IPF [6].

Surgical lung biopsy (SLB), which is recommended when no UIP pattern is present on the HRCT [3,7], is an invasive procedure that requires pleural drainage and is associated with a mortality rate ranging from 2.0% to 3.6% [8,9,10,11,12,13]. Moreover, in a recent study that included a cohort of patients with pathologically-proven UIP patterns, radiologists only identified a UIP pattern on thin-section CT with a sensitivity of 34% [14], according to the recent ATS-ERS guidelines [15]. Furthermore, the radiological assessment of fibrotic lung diseases is still challenging and often varies between experts [16,17,18,19]. Consequently, an automated approach that assists radiologists (especially less experienced ones) could be very useful in avoiding unnecessary biopsies in a context of a multidisciplinary discussion. 

The interest in radiomics, pioneered in 2012, has increased in recent years [20]. The field of handcrafted radiomics, briefly stated, extracting a large number of mineable quantitative data from medical images using predetermined formulas, has developed rapidly in recent times [20]. The term radiomics (handcrafted radiomics and deep learning) refers to the high-throughput extraction of numeric features from medical imaging modalities, providing high-dimensional data that could be used to identify patterns relating to the pathophysiology of a disease. These data could then be merged with the characteristics of each patient to aid clinical decision-making [20,21]. Different studies have shown that radiomics has the potential to complement clinical decision support systems, for example, for cancer diagnosis and prognosis [20,22,23,24]. These studies have shown some potential to function as imaging biomarkers and to predict clinical outcomes and drug responses [20,25,26,27]. While the potential of radiomics has mainly been investigated in oncology, it can also be applied to many other diseases, including ILDs and chronic obstructive pulmonary disease (COPD) [28,29,30].

We hypothesize that radiomic features are able to decode biological information from specified regions of interest within the lung that can be used to diagnose IPF with UIP pattern. The aims of this study are two-fold: (1) to evaluate the use of radiomics, to differentiate between normal lung tissue and ILDs; (2) to evaluate the use of radiomics to distinguish IPF with a typical or less typical (biopsy-proven) UIP pattern related to IPF from HRCT patterns related to non-IPF ILDs. We also conjecture, based on the literature [31], that tracheal enlargement and tracheal shape would significantly complement handcrafted radiomic features that would help in the classification of different types of ILDs.

## 2. Materials and Methods

### 2.1. Study Population

The study protocol was registered on Clinicaltrials.gov (identifier: NCT04430491), approved by the ethics committee of the Erasme University hospital (ref: P2017/411). The electronic medical records at Erasme University hospital (center i) were searched between 2011 and 2018 for patients diagnosed with ILD. The inclusion criteria were: (i) the availability of HRCT with slices of less than 1.5 mm; (ii) the availability of a high-confidence diagnosis (MDD diagnosis of IPF with a typical UIP pattern; MDD diagnosis of IPF with a biopsy-proven UIP pattern; or MDD diagnosis of non-IPF ILD, validated by a lung biopsy showing a pattern other than UIP). The exclusion criteria were (i) the use of contrast enhancements in HRCT; (ii) images containing metal or motion artifacts; and (iii) images reconstructed with a slice thickness larger than 1.5 mm (Figure 1). At least 1 chest physician, 1 pathologist, 1 thoracic radiologist, 1 specialist in internal medicine or rheumatology participated in the MDD. For external validation (database A), we used the group of patients diagnosed with interstitial lung diseases from the publicly available Lung Tissue Research Consortium (LTRC, https://ltrcpublic.com/ (accessed on 19 September 2018)). Images from patients with ostensibly healthy lungs (database B) were collected from the publicly available Radiomics Imaging Archive (RIA, https://www.radiomicsimagingarchive.eu/ (accessed on 24 October 2021)) (G4). Information was also gathered from patients, such as the demographic (age, gender) and clinical data (body mass index—BMI), as well as the measurements of pulmonary function tests (PFT) (forced expiratory volume in 1s (FEV1), forced vital capacity (FVC), and diffusion capacity of carbon monoxide (DLCO). The so-called gender, age, and pulmonary function (GAP) score and staging system that was developed by Ley et al. in 2012 [32] was calculated for each patient and the value was recorded.

### 2.2. High-Resolution CT (HRCT) Scanning

For center (i), the HRCTs were acquired on a 64- or 128-detector row CT system (Somatom, Definition, Siemens Healthineers, Erlangen, Germany). For database A, HRCT images were acquired using 4 different CT vendors (Siemens, Erlangen, Germany), (GE, Waukesha, WI, USA), (Philips, Amsterdam, the Netherlands), and (Toshiba, Tochigi-ken, Japan). For database B, all scans were acquired from the same scanner (GE Medical Systems, Waukesha, WI, USA). The slice thickness of all scans varied between 0.5 and 1.5 mm.

### 2.3. Segmentation

The process of delineating a region of interest (ROI) that will be utilized to extract handcrafted radiomic features is known as segmentation. A workflow for radiomics from segmentation to data analysis is depicted in Figure 2. Segmentation of the lungs and sectors, as well as the tracheobronchial tree, were performed automatically using an automated workflow created with MIM software (MIM Software Inc., Cleveland, OH, USA). Sectorized lung segmentation was performed to account for the differences in the spatial distribution of the lesions between UIP and non-UIP patterns. Each sector was defined as a (ROI). As shown in the left part of Figure 2, sectors 1 and 2 represent the upper section of the lung, sector 3 represents the middle section, and sector 4 represents the basal section.

### 2.4. Radiomic Features Extraction

To minimize the effects of the variations in image voxel size, all HRCT images were resampled into 1 × 1 × 1 mm^3^ voxel size, using linear interpolation to address the disparate reconstruction settings found in the datasets [33]. 1 × 1 × 1 mm^3^ was the maximum voxel size available in the dataset [34]. Radiomic features, except for the trachea volume, were extracted from the ROIs of the lung and sectors within the HRCT images, using the RadiomiX Discovery Toolbox (version, October 2019; https://www.radiomics.bio (accessed on 23 June 2020)), which calculates radiomics features in compliance with the Imaging Biomarkers Standardization Initiative (IBSI) [35]. Voxel intensities were aggregated into bins of 25 Hounsfield Units (HUs)—for nonfiltered features, excluding first-order statistics features—to reduce noise and interscanner variability [36]. The extracted features describe the fractal dimension, intensity histogram, first-order statistics, texture, and shape. Mathematical definitions and descriptions of the features mentioned can be found in other studies [21].

### 2.5. Data Splitting

For the first aim, i.e., normal vs. ILDs (G4 vs. G1,2,3), the data from center (i) and database B was combined and split into training and validation datasets, with a ratio of 0.8:0.2. For the second aim, i.e., IPF/UIP vs. non-IPF ILDs (G1 and 2 vs. G3), datasets from center (i) were randomly divided into training and validation dataset, using a ratio of 0.8:0.2, and data from database A was used as an external validation dataset.

### 2.6. Feature Selection and Modeling

To avoid any information leaking, all of the feature selection and model training was conducted in the training dataset alone. In order to reduce feature dimensionality, several steps were applied. Firstly, features with (near) zero variance (i.e., features that have the same value in ≥95% of the data points) were excluded. Next, feature pairs with Spearman correlation (r ≥ 0.90) were considered to be highly correlated, and the feature with the highest average correlation with all other features was removed. Then, the remaining features were fed into the Boruta dimension-reduction and feature-elimination algorithm, with the maximal number of important sources, runs set to 1000. The Boruta algorithm is a wrapper method based on random forest classification [37]. Afterward, a random forest model was trained with the remaining features and the top-10 features with the highest mean decrease in Gini were retained for the final random forest model. Five models were trained: 1 model was trained to classify between normal and ILDs, while the rest were used to classify between IPF with different UIP pattern appearances (i.e., UIP on HRCT or UIP not on HRCT but confirmed with a lung biopsy) and non-IPF ILDs with no UIP pattern and confirmed by a lung biopsy.

### 2.7. Statistical Analysis

All statistical analyses were performed using R on RStudio (version 4.0.2; https://www.R-project.org/ (accessed on 10 January 2022)). Comparisons between datasets were summarized using a Wilcoxon rank-sum test for the continuous variables and an X^2^ Fisher exact test for categorical variables. A Spearman correlation was used to evaluate the correlation between radiomic features.

To assess the model’s level of performance, the area under the curve (AUC) from the receiver operating characteristic (ROC) analysis was used and a 95% confidence interval (CI) was reported. To estimate the goodness-of-fit of the models, the Hosmer–Lemeshow test was used, and calibration plots were generated to visualize the consistency of models. This study was assessed using a Radiomics Quality Score [21] that consists of 16 items with different scores that sum up to 36 points and was designed specifically for radiomic studies.

## 3. Results

### 3.1. Patients Characteristics

A total of 328 patients were included in the study after the application of the exclusion criteria (Figure 1). A group of 122 patients from the center (i) was included. These patients were divided into three groups: (G1) patients with a final diagnosis of IPF and with typical UIP pattern in HRCT (n = 39); (G2) patients with non-typical UIP pattern and a final MDD diagnosis of IPF confirmed by SLB (n = 41); (G3) patients non-IPF ILD diagnosis confirmed by SLB (n = 42). From database (A), a total of 109 patients were included and divided into two groups: (1) IPF with UIP pattern patients (n = 53) and (2) non-IPF ILD with no UIP pattern (n = 56). From database (B) (G4), 97 healthy patients were included. A comparison between patients with a final diagnosis of IPF\UIP, non-IPF ILD, and healthy patients was performed and summarized in Table 1. As expected, there was a higher percentage of males among IPF patients (79% vs. 51%, *p* < 0.001), whereas no significant differences were noticed regarding age (*p* = 0.06), and lung function tests (FEV1, *p* = 0.8; FVC, *p* = 0.18; DLCO, *p* = 0.23; BMI, *p* = 0.34). 

### 3.2. Feature Extraction and Feature Selection

Original features were extracted (n = 170) for the whole and sectorized lung. Shape features and features with little or zero variance were excluded (n = 33). A list of the selected features after removing the highly correlated features, applying the Boruta algorithm, and Gini decrease can be found in Appendix A, Table A1. Feature selection methods yielded ten radiomics features as inputs for the group comparisons.

### 3.3. Performance of the Models

The volume of the trachea was observed to differ significantly (*p* < 0.001) between the control, IPF/UIP, and ILDs other than IPF patients (49.23 ± 12.96, 73.40 ± 22.01, and 61.67 ± 18.81 cm3, respectively, mean ± SD), and also between IPF/ UIP and ILD (non-IPF) (*p* < 0.001) (Figure 3). In addition, no association was detected between tracheal volume and either lung function (FVC% predicted, r = −0.03, *p* = 0.59), or the GAP index (r = 0.17, *p* = 0.01). Following the feature selection, the volume of the trachea was selected as an important feature for all models, except for the classification between normal and ILDs.

When classifying between a normal lung (G4, database B) and a lung with ILDs (G1 + G2 + G3) from center (i), an AUC of 1.0 (CI: 1.0–0.1) was achieved in validation (M1) (Figure 4). For the classification between G1 and G3 (center i), significant results were obtained using whole lungs with an AUC of 0.96 (95% CI: 0.90–1.0) in validation (M2). For the classification between G2 and G3 (center i), significant results were achieved using sector 1 (upper zone of the lung) with an AUC of 0.87 (95% CI: 0.74–1.0) in validation (M3).

When combining G1 and G2 to distinguish the results from G3 (center (i)), an AUC of 0.82 (95% CI: 0.68–0.95, M4) and 0.66 (95% CI: 0.59–0.73, M4.1) in validation and test dataset (database A) were achieved using whole lungs respectively. When 40% of the test dataset (from database A) is introduced to the training dataset, and retaining the remaining 60% as testing, an AUC of 0.77 (95% CI: 0.69–0.85) was achieved (M5).

The detailed sensitivity and specificity of the models for validation/testing dataset are summarized in Table 2. To gauge the presence of overfitting when retraining all the models with randomized outcomes, no single feature was chosen as significant when the Boruta algorithm was applied and the workflow had to be halted.

Among all models, M1, M2, and M4 showed proper calibration with *p* = 0.68, 0.32, and 0.07, respectively (Figure 5). The radiomics quality score of this study was 64% (23 of 36).

## 4. Discussion

In this study, we developed a quantitative signature (radiomics) extracted from HRCT to classify fibrotic lung disease. A random forest classifier was used to differentiate between (1) normal lungs and interstitial lung diseases (ILDs); (2) idiopathic pulmonary fibrosis (IPF) (with typical or less typical usual interstitial pneumonia (UIP) radiological presentation), and non-IPF ILDs (other than IPF as proven by the absence of UIP in a surgical biopsy). Briefly stated, we were able to demonstrate that radiomic features derived from HRCT images can be used to distinguish between a normal state and ILDs, as well as between IPF with a UIP pattern and ILDs with no UIP pattern verified by surgical biopsy. The inclusion of biopsy-proven non-IPF ILDs patients strengthens the study, as well as making it unique (Appendix A, Table A2).

Differentiating between normal and ILD lung tissues might seem a trivial task. However, it is a time-consuming process since the clinician has to go through all the scans. Developing an automated approach that differentiates between normal and abnormal lungs would decrease the amount of time a clinician needs to assess images on a daily basis. A previous study presented a novel texture analysis method that incorporates texture matching with histogram features analysis [38]. This study reported that their method achieved a sensitivity of 92.96% and a specificity of 93.78% in differentiating between normal and abnormal lungs. The study made use of a part of the handcrafted radiomic features used in our analysis. Using all-handcrafted radiomic features, we achieved a sensitivity of 98% and a specificity of 98% to identify an ILD.

Many ILDs have characteristics and changes in the lungs similar to those of IPF/UIP on HRCT, making the diagnosis very difficult—even for experienced radiologists [39]. Visual assessments of ILDs while using HRCT can be very subjective due to the high variability in the knowledge of inter-readers [16,17,18]. Therefore, providing automated diagnostic assistance in this setting would be highly beneficial, especially for less experienced radiologists. Texture image analysis is not new in fibrotic lung diseases and has been researched to automatically analyze ILDs on CT images [38,40,41,42,43,44,45,46]. However, most of the existing studies have focused on prognostic questions rather than providing diagnostic support. Maldonado et al. showed that short-term reticular changes evaluated by CALIPER (Computer-Aided Lung Informatics for Pathology Evaluation and Rating) correlated with physiological parameters and were predictive of survival in IPF patients [41]. Humphries et al. concluded that the use of Data-driven Texture Analysis (DTA) for IPF patients correlates with both pulmonary function tests and visual assessment on CT images at baseline [45]. However, a more thorough classification of phenotypes can be provided by applying radiomic data stratification. Walsh et al., used a deep learning approach for automated classification of fibrotic lung disease, according to the 2011 ATS/ERS/JRS/ALAT idiopathic pulmonary fibrosis diagnostic guidelines on a dataset of 1157 HRCT scans. The algorithm performance was compared to that of 91 radiologists and showed an accuracy of 73.3%, compared to the median accuracy of the radiologists, 70.7% [47]. To the best of our knowledge, no study has investigated the potential of handcrafted radiomics for differentiation between IPF/UIP and other ILDs.

By assessing the potential of handcrafted radiomics to differentiate between IPF with typical UIP presentation on HRCT and ILDs other than IPF, we discovered another benefit of automation similar to that achieved by differentiating between normal and abnormal lung tissue. It could serve mainly as a decision-aiding tool that would increase the diagnostic accuracy of the disease, reduce the need for invasive lung biopsies, and decrease the time needed to conduct routine scans.

IPF is also associated with wide parenchymal and airway conditions, such as those found in the trachea wall, which leads to pathological changes [48]. Ratwani et al., studied the correlation between the change of tracheobronchial tree size and the disease severity of IPF [31]. Our study found a significant difference in the volume of trachea between normal, IPF/UIP and, ILDs patients. Furthermore, it was found that the volume of the trachea was higher for IPF subjects compared to normal and ILDs other than IPF (Figure 3). No correlation was seen between the volume of the trachea and %FVC predicted. This conclusion may be consistent with the findings of Ratwani et al. [31], who found that there was no association between %FVC predicted and growing tracheobronchial tree size, indicating that tracheal expansion is not only due to fibrosis and that other variables may be at play. Such findings suggest that the increase of the volume of the trachea might be a good new handcrafted radiomic feature to serve as a promising tool in the diagnosis of IPF.

The decrease in model performance in the test dataset might be explained by the presence of variation in acquisition and reconstruction parameters. When the random forest algorithm learned part of the test dataset in the training dataset (M4.1), the model AUC increased from 0.66 to 0.77. Such findings indicate the need for addressing the challenges associated with differences in imaging parameters.

This study has some limitations. Firstly, we did have the additional categories of UIP patterns (definite, probable, indeterminate, or alternative) in the training dataset but not in the test dataset. Therefore, we only used the test dataset when we combined G1 and G2. Secondly, the healthy CT scans (G4) were obtained only from one center (center iii). Thirdly, the CT acquisition parameters of HRCT varied between and within the centers, and radiomic features are known to be influenced by different CT acquisition and reconstruction parameters [34,49,50]. Furthermore, we could not assess the reproducibility of features due to the lack of anthropomorphic phantom or test-retest scans acquired with settings similar to the scans used in this study. Henceforth, future studies must employ reproducibility studies to ensure the generalizability of the developed models. The application of radiomics to IPF may be broadened to include treatment decision aids. Further research should be undertaken to investigate the progression of IPF/UIP at baseline and follow up to evaluate the effectiveness of the antifibrotic treatment. In addition, a combination of deep learning and handcrafted radiomics with the addition of blood or genetic biomarkers would be a powerful tool in the classification of ILDs.

## 5. Conclusions

At present, there is minimal radiomics research on ILDs. Our findings are, nonetheless, promising and underline the strong potential of HRCT-based radiomics for the identification of ILDs. The classification between IPF/UIP and other ILDs using radiomics might capture features indicating different types of ILDs in HRCT, which are hardly recognizable via visual assessment. The radiomic features extracted from HRCT, along with clinical features, might aid in the assessment of ILDs and be used as a valuable tool for computer-aided decision-making in imaging.

## Figures and Tables

**Figure 1 jpm-12-00373-f001:**
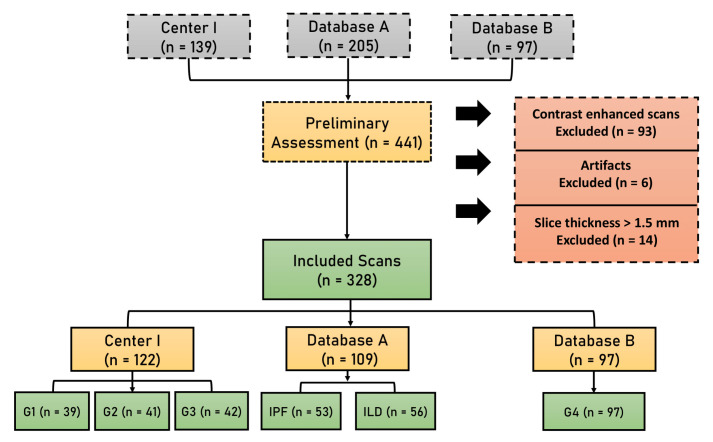
A flowchart diagram shows the patient selection process. (G1) patients with final MDD diagnosis of IPF with typical UIP pattern in HRCT and no lung biopsy; (G2) patients with a final MDD diagnosis of IPF confirmed by Surgical Lung Biopsy (SLB) (less typical HRCT pattern); (G3) patients with ILDs other than IPF with lung biopsy confirming a non-UIP pattern; (G4) patients with apparently healthy lungs.

**Figure 2 jpm-12-00373-f002:**
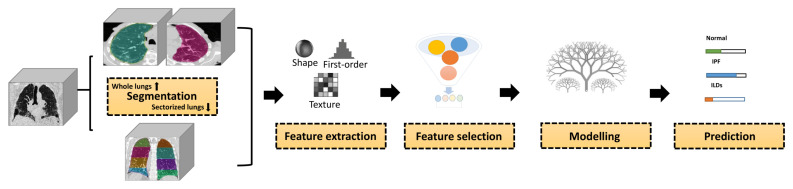
Radiomics Pipeline for lung fibrosis classification from CT images. First, the region of interest (ROI) was delineated. Second, handcrafted radiomic features were extracted from both ROIs. Third, feature selection methods were applied to select the most informative set of features. Fourth, the selected set of features were train the Random Forest classifier to arrive at a prediction.

**Figure 3 jpm-12-00373-f003:**
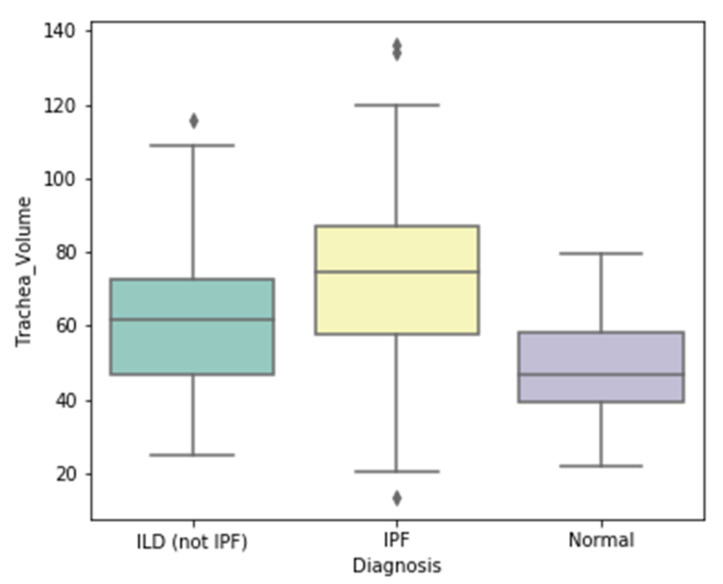
The difference in the volume of the trachea between IPF, non-IPF ILD, and normal, *p* < 0.001.

**Figure 4 jpm-12-00373-f004:**
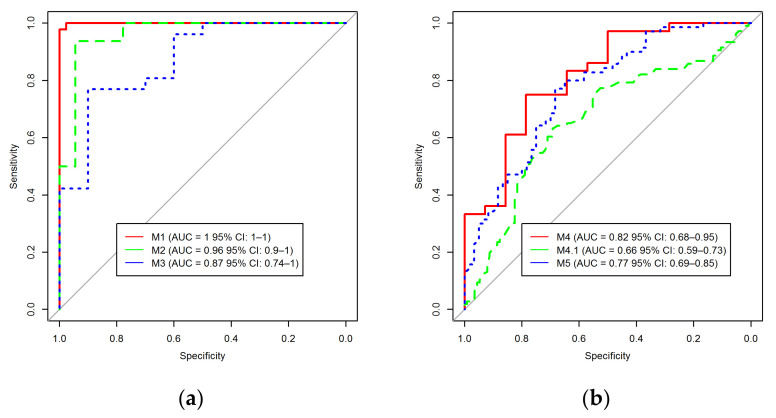
The graph shows the area under the receiver operating characteristic (AUC) curve of different models in the validation (**a**)\test (**b**) dataset. (M1) normal lungs vs. ILD; (M2) IPF\UIP on HRCT (G1) vs. non-IPF ILD (biopsy-proven) (G3); (M3) IPF\UIP pattern proven by biopsy (G2) vs. non-IPF ILD (biopsy-proven) (G3); (M4) IPF with UIP (G1 + G2) vs. non-IPF ILD (biopsy-proven) (G3); M4.1) IPF with UIP (G1 + G2) vs. non-IPF ILD (G3) vs. non-IPF ILD (biopsy-proven)(G3) in testing; (M5) IPF with UIP (G1 + G2) vs. non-IPF ILD (biopsy-proven) (G3) mixed with 40% of the testing dataset.

**Figure 5 jpm-12-00373-f005:**
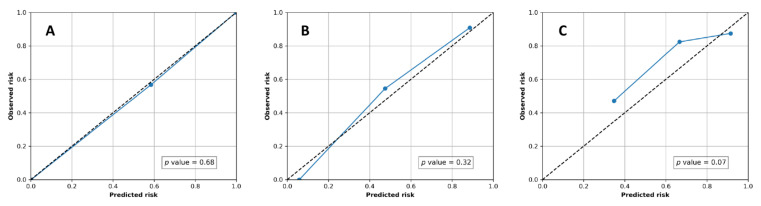
Calibration plots of radiomics models on the validation/testing dataset. (**A**) Normal vs. ILD (M1); (**B**) IPF\UIP vs. non-IPF ILD (M2); (**C**) IPF with UIP (G1 + G2) vs. non-IPF ILD (biopsy-proven) (M4).

**Table 1 jpm-12-00373-t001:** Demographic and clinical characteristics of patients with IPF, non-IPF ILD, and healthy groups. IQR: interquartile range; SD: standard deviation.

Variable	IPF\UIP (HRCT & Biopsy)	Non-IPF ILD (Biopsy)	Normal	*p*-Value
Age (median (IQR)	65 (60, 71)	63 (57, 72)	62 (56, 67)	0.06
Sex = M (%)	104 (78.8)	51 (51.5)	56 (57.7)	<0.001
FEV1 (mean (SD))	71.08 (18.34)	71.77 (21.94)	-	0.8
FVC (mean (SD))	67.39 (19.53)	71.07 (22.17)	-	0.18
DLCO (mean (SD))	38.92 (11.62)	36.73 (16.12)	-	0.23
BMI (mean (SD))	28.06 (4.42)	28.69 (5.59)	-	0.34

**Table 2 jpm-12-00373-t002:** Detailed predictive and diagnostic values among various models studied, using the validation/testing dataset.

Model (M)	AUC	Accuracy	Sensitivity	Specificity
	(95% CI)	%	%	%
M1	1.0 (1.0–1.0)	99	98	98
M2	0.96 (0.90–1.0)	91	88	94
M3	0.87 (0.74–1.0)	72	65	90
M4	0.82 (0.68–0.95)	70	66	79
M4.1	0.66 (0.59–0.73)	65	60	69
M5	0.77 (0.69–0.85	69	64	75

## Data Availability

The data from center (i) is privately owned while data from database A and B are publicly available.

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
