# Peer review of "A Handcrafted Radiomics-Based Model for the Diagnosis of Usual Interstitial Pneumonia in Patients with Idiopathic Pulmonary Fibrosis"

_jpm, 2022, doi:10.3390/jpm12030373_

Round 1

Reviewer 1 Report

Turkey Refaee and co-workers developed a quantitative signature extracted from HRCT to classify fibrotic lung diseases. The manuscript is original and little literature data is available about the investigation of EVs in IPF. The statistical analysis was appropriate and complete. Figures and tables were appropriate.  Just making footnotes more descriptive

Author Response

Dear reviewer,

We thank the reviewer for the praise and comment. Please see the attachment in the box.

Best wishes

Reviewer 2 Report

I read the article and have minor comments that are listed below:

Line 50: "ccording to these recommendations", it would be helpful to mention what are they are for both diagnostic and mangement.
Line 122: Good job sorting off figure 1, it is easy to follow.
Line 144: If you can define what segmentation is and the goal of it, would be benificary to the reader.
Line 152: Figure 2, While the 3D segmented image is good, if you could add its3D image with no segmentation to the figure, would illstrate the point.
Line 153: The title does not say much, you may edit more informaton to the reader.Most of the time, the reader likes to spend more time understanding the figure rather than reading the paper.
Line 185: Grammer: the remaining features" replaced with "the emaining features,"
Line 194: Grammer:"Spearman correlation was us" replaced with "A Spearman correlation was us"
Line 207: Grammer: "A group of 122 patients from center" replaced with "A group of 122 patients from the center"
Line 224: the table values are not displiyed within the tabble. Need further editting.
Line 332: The arrow at (M1), (M2), can be removed.. to M1, M2, etc..

Author Response

Dear reviewer,

We thank the reviewer for the praise and comment. Please see the attachment in the box. 

Best wishes, 
